# Facility readiness for decentralized Package of Essential Noncommunicable Disease Interventions-Plus (PEN-Plus) care in nine lower-income countries

Laura Drown[1‡], Alma J. Adler[1,2*‡], Devashri Salvi[1], Chantelle Boudreaux[1], Neil Gupta[1], Zipporah Ali[3], Neusa Bay[4], Mukobe Chisunka[5], Bavin Mulenga[5], Bhagawan Koirala[6], Bishwash Maharjan[6], Giacomo Marro[7], Zelalem Mengistu[8], Esther Mtumbuka[9], Martha Nabadda[10], Laura Ruckstuhl[11], Ian Wurie[12], Ada Thapa[1], Ana Mocumbi[4‡], Gene Bukhman[1,2‡], Emily B. Wroe[1,13‡], On Behalf of the PEN-Plus Working Group

1 Department of Medicine, Center for Integration Science in Global Health Equity, Brigham and Women's Hospital, Boston, Massachusetts, United States of America, 2 Department of Global Health and Social Medicine, Program in Global Noncommunicable Disease and Social Change, Harvard Medical School, Boston, Massachusetts, United States of America, 3 Noncommunicable Diseases Alliance Kenya, Nairobi, Kenya, 4 Universidade Eduardo Mondlane, Maputo, Mozambique, 5 Centre for Infectious Diseases Research in Zambia, Lusaka, Zambia, 6 Kathmandu Institute of Child Health, Kathmandu, Nepal, 7 Doctors With Africa CUAMM, Padova, Italy, 8 Mathiwos Wondu-Ye Ethiopia Cancer Society, Addis Ababa, Ethiopia, 9 Clinton Health Access Initiative, Dar es Salaam, Tanzania, 10 Uganda Initiative for Integrated Management of Non-Communicable Diseases, Kampala, Uganda, 11 SolidarMed, Harare, Zimbabwe, 12 Pujehun Hospital, Ministry of Health, Pujehun, Sierra Leone, 13 Partners In Health, Boston, Massachusetts, United States of America

‡ Laura Drown and Alma J. Adler share first authorship on this work. Ana Mocumbi, Gene Bukhman and Emily Wroe are joint senior authors on this work.
* Aadler2@bwh.harvard.edu

## Abstract

Severe chronic noncommunicable diseases (SC-NCDs) are important causes of avoidable disease burden in low- and lower-middle income countries (LLMICs) where care is often available only at tertiary, urban facilities. The Package of Essential Noncommunicable Disease Interventions – Plus (PEN-Plus) strategy aims to address gaps in access to care for SC-NCDs by integrating and decentralizing care. This study aims to assess baseline readiness of 16 facilities in nine LLMICs to provide care for SC-NCDs as part of a mixed-methods evaluation of PEN-Plus implementation. Cross-sectional surveys were utilized to collect baseline data from 16 facilities initiating new PEN-Plus programs. These surveys assessed the state of facility infrastructure and the availability of equipment and medicines for three priority conditions (type 1 diabetes (T1D), sickle cell disease (SCD), severe cardiac conditions (SCC)). Analysis consisted of descriptive statistics and summary index scores based on availability of key items. Facilities reported a high baseline availability of basic infrastructure. Readiness to provide care for priority SC-NCDs varied. Availability of functional diagnostic and management equipment and supplies for T1D was especially low in many facilities but higher for SCD. Medicine availability was overall highest for T1D (75%) but lower for SCD (39%) and SCC (49%), with significant gaps in essential medicines including hydroxyurea, anticoagulants, and medium- or long-lasting

**Data availability statement:** This is uploaded as supplementary information.

**Funding:** This work was supported by The Leona M. and Harry B. Helmsley Charitable Trust [2206-05274] to G.B. The funders had no role in the study design, data collection, analysis, decision to publish, or preparation of the manuscript.

**Competing interests:** The authors have declared that no competing interests exist.

insulins.These findings highlight the need for tailored, context-driven implementation approaches to address gaps in readiness for SC-NCD care in LLMICs. Baseline results will guide ongoing implementation and evaluation of the PEN-Plus clinics.

# Introduction

## Background

Noncommunicable diseases (NCDs) are a notable cause of death and disability among the world's poorest billion people, representing a growing proportion of the burden of disease in low- and lower middle-income countries (LLMICs) [1,2]. In this setting, much of this burden is caused by conditions we refer to here as severe chronic NCDs (SC-NCDs), including type 1 diabetes (T1D), sickle cell disease (SCD), and severe cardiac conditions (SCC), in addition to those typically addressed by global '5 x 5' NCD frameworks (cardiovascular disease, cancer, diabetes, chronic respiratory diseases, and mental health conditions) targeted by the WHO's Package of Essential Noncommunicable Disease Interventions (PEN). SC-NCDs account for more than a third of the total burden to disease among those living in extreme poverty and, in contrast to more common NCDs of the 5 x 5 framework, with an outsized proportion concentrated among individuals under 40 [1,3]. Given that many lower-income countries governments and global health donors have historically prioritized infectious and maternal diseases, readiness to diagnose and manage these complex SC-NCDs is often low, particularly outside of tertiary, urban settings [4,5]. Bridging this gap and expanding access to care for SC-NCDs among the poorest billion is a key priority of the NCDI Poverty Network [6].

## PEN-plus

The Package of Essential Noncommunicable Disease Interventions – Plus (PEN-Plus) is a strategy to build upon WHO PEN and address this unmet need and disease burden by decentralizing care for SC-NCDs in resource-constrained health systems [5,7]. It was first developed in Rwanda in 2006 then later scaled nationally in Rwanda starting in 2015 [8,9]. It has been implemented at facilities in Haiti, Malawi, and Liberia since 2018 [10,11]. In 2022, all 47 members of the World Health Organization (WHO) African Region adopted a resolution to initiate and expand PEN-Plus programs over the coming decade [12,13].

While the specifics of the program vary to reflect underlying epidemiologic needs, monitoring of the program is oriented around three sentinel conditions: type 1 diabetes, severe cardiac conditions, and (where relevant) sickle cell disease. These conditions have been identified to reflect both the large burden of disease and broad scope of clinical services required to address them [5,12]. These conditions require services such as insulin management, echocardiography, and prescription of medications for severe cardiac conditions and sickle cell disease that are often only available at tertiary level facilities in LLMICs. To improve access, PEN-Plus enables midlevel providers, including nurses, at intermediate level health facilities like district hospitals to provide such services as part of integrated care for a set of conditions.

Now, as part of a multi-country, three-year project supported by the NCDI Poverty Network, nine additional countries and one state in India are initiating PEN-Plus in a total of eighteen health facilities. The goals of these PEN-Plus initiation programs are to (1) provide high-quality service to 500–1000 patients with SC-NCDs per facility; (2) prepare providers at these facilities to train others in their region; and (3) support national health ministries to plan for PEN-Plus scale-up.

Participating hospitals vary greatly in existing service delivery, staffing, and overall systems at baseline and implementation approaches vary. However, all sites share operational plans and technical resources for PEN-Plus implementation, which involves training of midlevel providers and ensuring availability of equipment and medicines for appropriate diagnosis and treatment of SC-NCDs [5]. Establishment of PEN-Plus clinics at these facilities entails: 1) hiring, supervising, and supporting clinical and auxiliary staff, 2) staff training and mentorship, 3) procurement and distribution of supplies and commodities with supply chain support as needed, 4) supporting infrastructure for clinic space, and 5) systems development and support [14].

## PEN-Plus evaluation

As part of PEN-Plus initiation we have designed a cross-sectional multi-faceted evaluation of the program. This evaluation consists of multiple activities including a facility baseline assessment conducted at the start of the program, prior to PEN-Plus clinic opening, and repeated approximately two years into the program. Other activities include retrospective record reviews to be conducted one year and two years into implementation, simple costing activities, and interviews with policy makers at baseline and the end of the project. The overall evaluation is reported in Adler et al [14].

## Objectives

As part of the overall evaluation, we aim to understand the evolution of clinic space, staffing, medicine and equipment availability and the clinics' adequacy to become a training facility. To answer this question, we are tracking the progression of these items over baseline and endline. Here we present the outcomes of the baseline survey in 16 facilities in nine countries. For ease we are providing the data in three separate papers: overall readiness presented here, and two companion papers which focus on health information systems and health services organization. In this paper, we aim to understand and summarize facility readiness for provision of care for SC-NCDs at baseline prior to PEN-Plus initiation in 16 clinics.

## Materials and methods

### Study design

We developed two surveys (Part A and Part B [S1 Appendix]) as part of this baseline assessment. Part A of the survey consisted of a facility-wide assessment, including questions on background, infrastructure, service lines/departments present at the facility, conditions managed, staffing, diagnostics, medications, and equipment. Part B of the survey was more tailored to NCD specific management and health information data systems. The survey was developed in a collaborative process beginning in November 2021, building on our team's previous work analyzing equipment and medication availability for NCDs [15,16]. Survey design was guided by strategic information needed to inform PEN-Plus initiation at the selected clinics. The survey underwent multiple iterative rounds of development where the initial draft was created by the research team and clinicians provided input throughout the development process. After refinement of an initial survey, we consulted with partners in each country to ensure the tool was appropriate to the target settings. Once finalized, the tool was then distributed to the sites in two formats: electronically through REDCap [17,18] and as a paper- based to accommodate varying levels of technological access.

### Setting

**Site selection.** National NCDI Poverty Commissions established in 23 LLMICs, primarily those in which the world's poorest billion reside, have assessed the national burden of

NCDs and injuries and prioritized policies, interventions, and integrated delivery models such as PEN-Plus to address the disease burden [19]. Countries that completed a National NCDI Poverty Commission were invited to submit a letter of interest in becoming a PEN-Plus training site in July of 2020 [20]. Ministry of Health support was a requirement for all submissions. Final clinical sites – generally first-level hospitals such as district or community hospitals – were selected by the Ministries of Health in consultation with the implementing partners and the NCDI Poverty Network. Between June - August 2021, we selected 12 implementing partners from 10countries. The 18 clinics chosen for PEN-Plus initiation have varying models of service delivery, training, and overall system. Sixteen clinics in nine countries are represented in this paper (Table 1) – two clinics in one country opted out of participation in this publication in accordance with national guidelines on data sharing.

## Survey implementation

The finalized survey was distributed to all selected sites in March 2022 and they were given the opportunity to complete the survey via paper forms or electronic entry using REDCap (Research Electronic Data Capture) [17,18]. Electronic surveys were built in REDCap and members from all country teams were provided with training sessions with research team staff conducted over Zoom. These sessions provided detailed walkthroughs of the survey structure, REDCap interface, and instructions for completing the surveys. Virtual office hours were held for the countries to address the question regarding survey completion and data input. Training sessions were provided by a researcher with experience in survey administration and familiarity with the PEN-plus initiative. Four countries elected to complete the surveys electronically using REDCap following a virtual training session, while the remaining completed paper forms. Paper surveys were manually entered

**Table 1. Participating countries, implementing partners, and clinics.**

| Country | Implementing Partner(s) | Country Facility Number | Facility Type | Facility Setting | Approximate Catchment Population |
|---|---|---|---|---|---|
| Ethiopia | Mathiwos Wondu Ye - Ethiopia Cancer Society | 1 | Public | Peri-urban | 305,000 |
| | | 2 | Public | Peri-urban | 500,000 |
| Kenya | NCD Alliance Kenya | 1 | Public | Rural | 176,000 |
| | | 2 | Public | Rural | 116,000 |
| Mozambique | Universidade Eduardo Mondlane Instituto Nacional de Saúde Doctors With Africa CUAMM Mozambique Institute for Health Education and Research | 1 | Public | Rural | 318,000 |
| Nepal | Kathmandu Institute of Child Health | 1 | Public | Urban | 427,000 |
| | | 2 | Public | Urban | 300,000 |
| Sierra Leone | Partners In Health | 1 | Public | Rural | 621,000 |
| | Doctors With Africa CUAMM | 2 | Public | Rural | 407,000 |
| Tanzania | National Institute for Medical Research | 1 | Public | Rural | 245,000 |
| Uganda | Uganda Initiative for Integrated Management of Non-Communicable Diseases | 1 | Public | Rural | 300,000 |
| | | 2 | Public | Rural | 301,000 |
| Zambia | Centre for Infectious Diseases Research in Zambia | 1 | Public | Rural | 421,000 |
| | | 2 | Public | Urban | 479,000 |
| Zimbabwe | SolidarMed Clinton Health Access Initiative | 1 | Public | Rural | 195,000 |
| | | 2 | Public | Rural | 174,000 |
| Total | – | 16 | – | – | 5,285,000 |

into REDCap by Boston-based staff. Most countries administered the survey in English, as originally written, though one translated it into the local language (Portuguese). The translation process involved collaboration with bilingual team members familiar with technical terminology used in the survey. We received responses from all facilities over the period of June 2022 to February 2023. Responses represented status of the facilities at the time of survey submission. No human subjects were included in this research.

All surveys were checked for completeness and consistency centrally by the research team. Sites were contacted and resent previous surveys to complete and/or correct responses that were incomplete or unclear. In some instances, we asked sites to confirm or clarify responses. This process was repeated as needed until surveys were complete.

## Analysis

Data collected by the baseline assessment for each site featured a wide range of variables pertaining to broader domains. In this paper, we utilized a subset of variables related to 1) facility background, 2) facility infrastructure, 3) availability and functionality of diagnostic and management equipment and supplies, and 4) availability of medicines.

Analysis was conducted at the clinic level. It consisted of simple descriptive statistics (numbers and percentages of total) for variables pertaining to facility background and infrastructure. For the key conditions of interest (T1D, SCD, and SCC), we utilized index scores for 1) availability of functional diagnostic and management equipment and supplies at the time of the survey and 2) availability of medicines at the time of the survey. Scores for each facility were calculated on a range from zero to one hundred percent based on index components fulfilled (Table 2). Index components were included based on consultation with clinicians familiar with management of the three conditions. All index components were weighted equally, with the score calculated as the number of fulfilled components divided by the total number of components for that specific index, multiplied by one hundred. As SCD is not a priority condition in Ethiopia due to epidemiology, we did not calculate index scores for the two clinics located there.

As most clinics did not have a method of tracking patients with SC-NCDs at baseline, we were unable to include patient numbers from baseline surveys. In order to track numbers of people living with SC-NCDs over time, we included patient numbers from each clinics' first routine quarterly report.

## Ethics

This study was reviewed and determined to not be human subject's research by the Mass General Brigham IRB. This work was deemed exempt or covered by Mass General Brigham #2022P001390, Nepal Health Research Council 571/2022 P, National Institute for Medical Research (Tanzania) NIMR/Hq/R.8aVol.IX/4184, University of Zambia Biomedical Research Ethics Committee #3032–2022, Mulago Hospital Research and Ethics Committee MHREC 2022–74, Sierra Leone Ethics and Review Committee SLESRC 027/01/2025, Medical Research Council of Zimbabwe MRCZ/E/347, Amhara Public Health Institute NoH/R/T/T/D/07/43, and Oromia Health Care Bureau BF/HBTFH/H6/2027.

## Results

### Participants

All sixteen sites in nine countries submitted baseline assessments (S1 Data), representing a completion rate of 100%. Most sites (n = 11, 68.2%) were in rural areas (Table 3). The population served by these health facilities were predominantly rural (n = 14, 87.5%), though some

**Table 2. Index components by condition.**

| Condition | Index | Components |
|---|---|---|
| Type 1 diabetes | Diagnostic and management equipment and supplies | Laboratory-based blood glucose testing<br>Home glucometers<br>Hemoglobin A1c<br>Urine dipstick for ketone testing |
| | Medicines | Short-acting insulin<br>Intermediate-acting or long-acting insulin<br>Metformin<br>Sulfonylurea |
| Sickle cell disease | Diagnostic and management equipment and supplies | Hemoglobin testing<br>Full blood count<br>Sickle cell testing |
| | Medicines | Hydroxyurea<br>Prophylactic antibiotics |
| Severe cardiac conditions | Diagnostic and management equipment and supplies | Electrocardiography<br>Ultrasound equipment with cardiac probes<br>Blood pressure measuring devices<br>Weight scale<br>Serum electrolytes<br>Creatinine<br>Coagulation (prothrombin time (PT)/INR) |
| | Medicines | Aspirin<br>Loop diuretics<br>Angiotensin-converting enzyme inhibitors<br>Beta-blockers<br>Potassium-sparing diuretics<br>Calcium channel blockers<br>Methyldopa<br>Heparin<br>Warfarin<br>Other (non-warfarin) oral anti-coagulant<br>Potassium, oral<br>Hydralazine<br>Isosorbide dinitrate<br>Benzathine penicillin or penicillin V potassium |

reported serving urban areas (n = 7, 43.8%) and several reported serving both rural and urban populations. Nearly all (n = 15, 93.8%) facilities are owned and managed by governments.

## Facility infrastructure

Key infrastructure components are reported in Table 3. More than half of facilities (n = 11, 68.8%) reported always having electricity within the seven days prior to survey administration. Only one facility (6.3%) reported poor availability of electricity, with this facility responding that they never had electricity within the prior seven days. All facilities (100%) reported having water available on premises, an ambulance available on site or nearby, and a pharmacy on site. Availability of a functional telephone, internet access, and a functional computer was very high across sites (Table 3).

## Availability of diagnostic and management equipment and supplies

Sites reported availability of functional diagnostic and management equipment and supplies essential for care for three conditions: T1D, SCD, and SCC. The resulting index scores (Table 4) demonstrate large variability of availability across facilities as well as conditions. Facilities generally reported lower availability of diagnostic and management equipment and supplies

**Table 3. Basic characteristics and infrastructure of facilities.**

| Facility characteristic | Number of facilities (%) (N = 16) |
|---|---|
| Facility location | |
| Rural | 11 (68.18) |
| Peri-urban | 2 (12.5) |
| Urban | 3 (18.8) |
| Facility catchment area type | |
| Rural | 14 (87.5) |
| Urban/Peri-urban | 7 (43.8) |
| Managing authority | |
| Government | 16 (100.00) |
| **Infrastructure component** | |
| Electricity availability within 7 days prior to survey | |
| Always (no interruptions) | 11 (68.8) |
| Often (interruptions of <2 hours per day) | 4 (25.0) |
| Sometimes (frequent or prolonged interruptions of >2 hours per day) | 0 (0.0) |
| Rarely (intermittently or infrequently available) | 0 (0.0) |
| Never | 1 (6.3) |
| Water available on premises | 16 (100.0) |
| Ambulance available at or nearby facility | 16 (100.0) |
| Functional telephone | 13 (81.3) |
| Internet access | 14 (87.5) |
| Functional computer | 14 (87.5) |
| Pharmacy on site | 16 (100.0) |

for T1D compared with the other two conditions. For T1D, scores ranged from 0% with two facilities (12.5%) reporting having no availability of these functional items to an equal number receiving a score of 100%, indicating availability of all items. In this index, availability was lowest for hemoglobin A1c (HbA1c), with only three facilities (18.8%) reporting having functional equipment to conduct this test. Of the fourteen facilities reporting on SCD, nearly one-third (n = 5, 35.7%) reported having all three index components available and functional at the time of the assessment. The majority (n = 9, 64.3%) reported either one (n = 1, 7.1%) or two (n = 7, 50.0%) key components to be not present or not functional. Only half of facilities reported being able to provide diagnostic testing for SCD at the time of the survey. A longer list of inputs was included in the index for SCC. Only one facility (6.3%) had a perfect index score of 100% and, though over half of facilities had a score of less than 50% (n = 9, 56.3%), all facilities had at least one functional component. Availability was lowest for ultrasound equipment with cardiac probes for performing echocardiography – only one facility (6.3%) reported this to be available and functional.

## Availability of medicines

Index scores for availability of medicines for the three priority conditions were similarly varied (Table 5). Despite having a wide range of facility index scores, availability of medicines for T1D was overall quite high with a mean index score of 75%. Half of all sites reported all medicines included in this index to be available at the time of the baseline survey. For T1D, availability was lowest for intermediate or long-lasting insulin, which was lower (n = 9, 56.3%) than that of short-acting insulin (n = 12, 75.0%) Availability of medicines was considerably

**Table 4. Index (0 to 100%) of reported availability of functional diagnostic and management equipment and supplies for type 1 diabetes, sickle cell disease, and severe cardiac conditions by country and facility.**

| Condition | Country* | | | | | | | | | | | | | | | | | |
|---|---|---|---|---|---|---|---|---|---|---|---|---|---|---|---|---|---|---|
| | ETH | | KEN | | MOZ | | NEP | | SLE | | TAN | | UGA | | ZAM | | ZIM | |
| Type 1 diabetes | 50 | 50 | 25 | 0 | 0 | | 100 | 100 | 75 | 50 | 75 | 50 | 75 | 50 | 75 | | 75 | 25 |
| Sickle cell disease | N/A+ | N/A+ | 67 | 33 | 0 | 67 | 100 | 100 | 67 | 67 | 67 | 67 | 100 | 100 | 100 | | 67 | 67 |
| Severe cardiac conditions | 29 | 57 | 29 | 14 | 57 | | 100 | 086 | 43 | 29 | 29 | 43 | 71 | 57 | 43 | | 57 | 29 |

*ETH = Ethiopia, KEN = Kenya, MOZ = Mozambique, NEP = Nepal, SLE = Sierra Leone, TAN = Tanzania, UGA = Uganda, ZAM = Zambia, ZIM = Zimbabwe.

+SCD not condition of focus in Ethiopia due to epidemiology

Rose shades indicates 0 – 25%; Orange shades indicates 26 – 50%; Yellow shades indicates 51 – 75%; Green shades indicates 76 – 100%

**Table 5. Index (0 to 100%) of reported availability of medicines for type 1 diabetes, sickle cell disease, and severe cardiac conditions by country and facility.**

| Condition | Country* | | | | | | | | | | | | | | | | | |
|---|---|---|---|---|---|---|---|---|---|---|---|---|---|---|---|---|---|---|
| | ETH | | KEN | | MOZ | | NEP | | SLE | | TAN | | UGA | | ZAM | | ZIM | |
| Type 1 diabetes | 100 | 100 | 100 | 25 | 75 | | 100 | 75 | 0 | 50 | 75 | 100 | 50 | 50 | 100 | | 100 | 100 |
| Sickle cell disease | N/A+ | N/A+ | 0 | 0 | 50 | 50 | 100 | 0 | 50 | 0 | 50 | 100 | 0 | 50 | 50 | | 50 | |
| Severe cardiac conditions | 64 | 64 | 50 | 21 | 71 | | 64 | 71 | 0 | 57 | 50 | 57 | 57 | 50 | 36 | | 36 | 36 |

*ETH = Ethiopia, KEN = Kenya, MOZ = Mozambique, NEP = Nepal, SLE = Sierra Leone, TAN = Tanzania, UGA = Uganda, ZAM = Zambia, ZIM = Zimbabwe.

+SCD not condition of focus in Ethiopia due to epidemiology

Rose shades indicates 0 – 25%; Orange shades indicates 26 – 50%; Yellow shades indicates 51 – 75%; Green shades indicates 76 – 100%

lower for the other two conditions. Of the 14 sites focusing on SCD, half (n = 7, 50.0%) had only one medicine. Two (14.3%) had both essential medicines, and five (n = 4, 35.7%) had neither hydroxyurea nor prophylactic antibiotics. Only two (14.3%) facilities reported hydroxyurea to be available at the time of the survey. Index scores tracking the availability of the twelve medicines for SCC ranged widely from 7% at one (6.3%) site to 71% in two (12.5%). While all sites reported that at least one medicine in the index was available at the time of the survey, no facility had all fourteen. Availability was particularly low for anticoagulants – only two (12.5%) facilities reported heparin to be available, while warfarin was available in four (25.0%). Only one facility (6.3%) reported carrying another type of oral anticoagulant.

## Management of priority conditions at baseline

Within the first three months of beginning implementation, the majority of facilities (n = 10, 62.5%) reported caring for at least one patient with each priority condition. The number of patients enrolled in care varied widely by condition and facility (Table 6). Overall, SCD patients comprised the largest group, with over three-quarters (n = 731, 76.7%) of these patients enrolled at a single facility in Uganda. Four sites, excluding those in Ethiopia where it is not a condition of focus due to epidemiology, had not yet enrolled any SCD patients. In comparison, for T1D and SCC nearly all facilities (n = 15, 93.8%) had enrolled at least one patient.

## Discussion

### Summary of main results

We found that while the facilities initiating PEN-Plus overall reported strong basic facility infrastructure, readiness to provide care for T1D, SCD, and SCC as indicated by availability of functional diagnostic and management equipment and supplies, and medicines were

Table 6.  Number of patients with type 1 diabetes, sickle cell disease, and severe cardiac conditions in care at baseline by country and facility.

| Condition | Country* | | | | | | | | | | | | | | | |
| --- | --- | --- | --- | --- | --- | --- | --- | --- | --- | --- | --- | --- | --- | --- | --- | --- |
| | ETH | | KEN | | MOZ | | NEP | SLE | | TAN | UGA | | ZAM | | ZIM | |
| Type 1 diabetes | 166 | 42 | 1 | 5 | 6 | 1 | 0 | 41 | 6 | 5 | 60 | 17 | 6 | 22 | 17 | 2 |
| Sickle cell disease | N/A+ | N/A+ | 3 | 0 | 0 | 25 | 0 | 37 | 5 | 3 | 731 | 89 | 13 | 46 | 1 | 0 |
| Severe cardiac conditions | 16 | 12 | 4 | 0 | 7- | 1 | 10 | 116 | 4 | 2 | 4 | 1 | 2 | 63 | 38 | 5 |

*ETH = Ethiopia, KEN = Kenya, MOZ = Mozambique, NEP = Nepal, SLE = Sierra Leone, TAN = Tanzania, UGA = Uganda, ZAM = Zambia, ZIM = Zimbabwe.

+SCD not condition of focus in Ethiopia due to epidemiology

heterogenous between both facilities and conditions. Of the three conditions, readiness to diagnose and manage SC-NCDs as demonstrated by the mean index availability of functional diagnostic and management equipment and supplies was highest for SCD (72%) and fairly similar between T1D and SCC (55% and 48%, respectively). In contrast, readiness in terms of medicine availability was highest for T1D (mean: 75%), for which half of the facilities reported all index components were available at the time of the survey. Across all indices, scores ranged widely between facilities.

We identified key diagnostic and management equipment and supplies, and medicines for management of SC-NCDs that were particularly lacking in availability at these facilities. For T1D, few facilities reported being able to provide HbA1c testing. Availability of intermediate- and long-lasting insulin was also fairly low, limiting clinics' ability to provide basal bolus regimens per PEN-Plus standards of care. Only one site reported having a functional ultrasound with cardiac probes for echocardiography, an essential service for both initial diagnosis and routine follow-up of SCC. Low availability of anticoagulants is another limitation in the management of SCC at baseline in these clinics. Of the 14 clinics focusing on SCD, only half reported being able to provide diagnostic testing for this condition. Availability of hydroxyurea was low, with the medication available in only two clinics.

Patient enrollment at the PEN-Plus clinic at the time of the first routine quarterly report was variable across both facilities and the three priority conditions. While several facilities had not yet begun to enroll patients for all priority conditions, and numbers of enrolled patients were low for others, we anticipate these numbers will increase as facilities' readiness to diagnose and manage these conditions improves and the facilities begin case-finding activities.

## Comparison with existing literature

Literature has previously identified gaps in basic infrastructure at facilities in low- and middle-income countries, particularly related to availability of continuous electricity [21,22]. The percentage of facilities reporting electricity to be always available (68.8%) was slightly lower in this study than in district hospitals in Tanzania (87%) and Zambia (75%) participating in a study of anesthesia capacity, though higher than those in Malawi (23%) [22]. The facilities in this assessment reported higher availability of water compared with those participating in the anesthesia study.

Many of our findings aligned with a 2020 study utilizing Service Provision Assessment (SPA) data to determine availability and medications for NCDs at public, first-level hospitals in eight countries, including three countries participating here (Ethiopia, Nepal, and Tanzania) [15] Both this earlier study and the current assessment found low availability of medication and equipment for T1D and SCC at facilities, though the former did not include SCD. Another study using SPA data to assess capacity to diagnose and manage diabetic ketoacidosis (DKA) in nine countries also found overall low availability of essential items, including insulin [16] In the DKA study, availability of essential items also varied greatly by country as

well as facility level – for example, insulin availability ranged from 2% at secondary facilities in Bangladesh to 100% at tertiary ones in Malawi.

## Limitations

Since this survey only includes one to two facilities per country, it is not nationally representative or representative of other facility types in these countries. Despite this limitation, this data offers an understanding of the current status of high impact, underreported clinical services that is not easily gleaned from existing data. Across all countries, sites were selected to be characteristic of first-referral-level care in facilities serving primarily rural populations rather than spanning health system levels. Since they were selected as part of a scale-up strategy, however, they may be better equipped than other rural first-referral level facilities. Importantly, this data is self-reported by participating facilities and while data was checked for completeness and content centrally, it was not externally validated. Some data points, such as availability of an ambulance, are summaries of overall self-reported impressions so while reported as available may experience some inconsistency or lack of functionality at times, which may not be captured in this data. All medications and diagnostics included in indices are reported with equal weight, which may not fully capture clinically significant shortages. This methodology may under- or overemphasize the impact of a given input relative to others, but it  enables us to provide a rapid snapshot of overall readiness to deliver care for the three conditions. Another potential limitation of thesedata in relation to the overall evaluation is the timing of its collection. Due to the process of following up with facilities to collect missing data as well as logistical delays in data collection for some sites, it is possible some data points may over-estimate baseline availability. However, we will still be able to track progress of the facilities over the course of project implementation as intended.

## Implications and recommendations

This assessment provides us with a snapshot of readiness across 16 diverse first-level hospitals across sub-Saharan Africa and South Asia. These results demonstrate the need for understanding local conditions and context-specific needs when initiating NCD care provision models such as PEN-Plus. Implementation of expanded baseline assessments to understand existing areas of strength and weakness in the health system allows for targeted, more effective implementation approaches to health system strengthening. Here we found gaps in availability of equipment for diagnosis, ongoing monitoring, and medication across conditions. In T1D, however, gaps in diagnostic and monitoring equipment are especially concerning, given the dangers of insulin administration. These findings highlight the importance of robust referral pathways to supplement primary care level services for management of NCDs. At the first point of referral – the district-level hospital in many countries – it is essential to ensure capacity of emergency and inpatient care for diagnosis, as well as the monitoring systems needed for longitudinal outpatient care.

In integrated models, identifying and leveraging complementarities in care allows the delivery of a number of well-aligned services utilizing the same space and staff. The findings presented here provide a snapshot of facility readiness prior to the implementation of PEN-Plus.

The gaps identified in this analysis will be addressed during PEN-Plus initiation, with the exception of those in basic infrastructure such as electricity and water supply. Thus, we anticipate an improvement in readiness at these sites to provide care for all three key conditions related to availability of diagnostic and management equipment and supplies, and medicines. At baseline, the sixteen facilities reported differing readiness to manage T1D, SCD, and SCC

as represented by the six index scores (availability of functional diagnostic and management equipment and supplies and availability of medicines for each condition). We expect that this heterogeneity will be reflected in the implementation of PEN-Plus clinics. For example, implementation costs and rates at which each clinic is able to manage patients with SC-NCDs will depend on their health systems at baseline. Similarly, this diversity will impact the evaluation of this program. As health systems of the facilities vary at baseline, we do not expect progress in readiness to manage these conditions to necessarily be equal across facilities in the subsequent endline evaluation. Therefore, we will compare these sites only to themselves across the course of the study to measure progress.

Finally, we urge others considering implementation of PEN-Plus or similar integrated care programs to adapt implementation based on local epidemiology, context, and needs.

## Conclusion

This assessment demonstrates that readiness for provision of care for SC-NCDs at baseline is variable at facilities initiating PEN-Plus, reflecting the diversity of implementation settings at initiation of this project. Such heterogeneity will guide our approaches to both implementation and evaluation of PEN-Plus clinics. Given the gaps in key equipment and supplies needed to diagnose and manage SC-NCDs at these facilities, strengthening supply chain systems will be a critical component t of PEN-Plus implementation. We will monitor progress towards provision of integrated care for SC-NCDs at these facilities and report on developments in readiness at the endline survey in subsequent papers.

## Supporting information

**S1 Text.  PEN-Plus Partnership collaborators** .
(DOCX)

**S1 Data.  Data included in the analysis** .
(PDF)

**S1 Appendix.  PEN-Plus Partnership Baseline Survey** .
(PDF)

## Acknowledgement

We are grateful to all the providers working in the PEN-Plus clinics.

## Author contributions

**Conceptualization:** Laura Drown, Alma Adler, Chantelle Boudreaux, Ana Mocumbi, Gene Bukhman, Emily B. Wroe.

**Data curation:** Devashri Salvi.

**Formal analysis:** Laura Drown, Alma Adler, Devashri Salvi, Chantelle Boudreaux.

**Project administration:** Laura Drown.

**Supervision:** Ana Mocumbi, Gene Bukhman, Emily B. Wroe.

**Writing – original draft:** Laura Drown, Alma Adler.

**Writing – review & editing:** Laura Drown, Alma Adler, Devashri Salvi, Chantelle Boudreaux, Neil Gupta, Zipporah Ali, Neusa Bay, Mukobe Chisunka, Bavin Mulenga, Bhagawan Koirala, Bishwash Maharjan, Giacomo Marro, Zelalem Mengistu, Esther Mtumbuka,

Martha Nabadda, Laura Ruckstuhl, Ian Wurie, Ada Thapa, Ana Mocumbi, Gene Bukhman, Emily B. Wroe.

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
