## [Decision Letter · Decision Letter 0]

3 Jan 2025

PGPH-D-24-02559

Facility readiness for decentralized PEN-Plus care for severe chronic non-communicable diseases in nine lower-income countries

Dear Dr. Adler,

Thank you for submitting your manuscript to PLOS Global Public Health. After careful consideration, we feel that it has merit but does not fully meet PLOS Global Public Health’s publication criteria as it currently stands. Therefore, we invite you to submit a revised version of the manuscript that addresses the points raised during the review process.

We look forward to receiving your revised manuscript.

Kind regards,

Joel Msafiri Francis, MD, MS, PhD

Academic Editor

Journal Requirements:

Additional Editor Comments (if provided):

Reviewers' comments:

Reviewer's Responses to Questions

**Comments to the Author**

1. Does this manuscript meet PLOS Global Public Health’s publication criteria ? Is the manuscript technically sound, and do the data support the conclusions? The manuscript must describe methodologically and ethically rigorous research with conclusions that are appropriately drawn based on the data presented.

Reviewer #1: Yes

Reviewer #2: Yes

2. Has the statistical analysis been performed appropriately and rigorously?

Reviewer #1: I don't know

Reviewer #2: Yes

3. Have the authors made all data underlying the findings in their manuscript fully available (please refer to the Data Availability Statement at the start of the manuscript PDF file)?

Reviewer #1: No

Reviewer #2: Yes

4. Is the manuscript presented in an intelligible fashion and written in standard English?

Reviewer #1: Yes

Reviewer #2: Yes

5. Review Comments to the Author

Reviewer #1: The article meets the criteria for PLOS Global public health journal publication and has a sound manuscript with conclusions that are supported by data presented in the results section. The design for the primary research question posed projects an ethically and methodologically sound design with mentioned limitations and weaknesses. This is further elaborated by the drawn conclusions which even with the mentioned limitations in the design, not only answer the question posed but go on to approximate the readiness of 16 centers in 9 low to middle income countries towards 3 severe chronic non communicable diseases.

However, data availability seems inadequate as some of the procedures and surveys implemented are not fully detailed. The manuscript is presented in good palatable english

Some recommendations;

Provide the full meaning of PEN-plus in the title to provide transparency

Line 65- according to WHO (https://www.who.int/news-room/fact-sheets/detail/noncommunicable-diseases & https://www.afro.who.int/health-topics/noncommunicable-diseases), respiratory conditions, cancers, rheumatic conditions and Hypertension are also conditions we should be ready for so to strengthen your 3 choices you need a substantial literature review supporting why you chose the 3 you chose for the 9 countries selected.

Line 66- grammar error (some text missing)

Line 118/119- survey design is mentioned but not fully described, to support reproducibility of results all data need to be available.

Line 141- provide more details on the survey, the questionnaire, language used, qualifications of trainers, assessments of understanding and what criteria justified clarifying or confirming responses etc

Line 162/163 what time point was the survey conducted because given the time range of june 2022-february 2023, its not clear which time point was the data collected to assess the availability of medicines, equipment etc.

Reviewer #2: Reviewer comments

Topic: Facility readiness for decentralized PEN-Plus care for severe chronic noncommunicable diseases in nine lower-income countries

Summary: The study assessed baseline facility readiness of 16 selected clinics in nine low-income countries as part of a mixed-methods evaluation of the implementation of the PEN-PLUS strategy, which aims to address gaps in access to care for SC-NCDs by integrating and decentralizing care.

Here are a few comments to improve the manuscript:

Abstract

1) Line 47: Not sure whether it is “hospitals” or clinics”, please correct accordingly

2) Line 55: The interpretation of index scores (0.75, 0.39, 0.49) is not clear from the abstract. The authors may add an explanation or maybe put in percent.

Introduction

1) Line 66: cross out “than”

2) Line 85: add a comma, before “, nine”

Materials and methods

1) Line 163-4; For ease of interpretation, I would suggest converting the index scores to percentages, by multiplying by 100.

2) Line 175-6: More details on ethics considerations? Was ethics sought in each country? Through what Ethics Committees/Institutions? Anonymity?

Results

1) No further comments

Discussion

1) Line 282: cross out the first “in”

6. PLOS authors have the option to publish the peer review history of their article (what does this mean? ). If published, this will include your full peer review and any attached files.

**Do you want your identity to be public for this peer review?** For information about this choice, including consent withdrawal, please see our Privacy Policy .

Reviewer #1: No

Reviewer #2: No

---

## [Editor Report · Decision Letter 1]

25 Feb 2025

Facility readiness for decentralized Package of Essential Noncommunicable Disease Interventions-Plus (PEN-Plus) care in nine lower-income countries

PGPH-D-24-02559R1

Dear Dr Adler,

We are pleased to inform you that your manuscript 'Facility readiness for decentralized Package of Essential Noncommunicable Disease Interventions-Plus (PEN-Plus) care in nine lower-income countries' has been provisionally accepted for publication in PLOS Global Public Health.

Best regards,

Joel Msafiri Francis, MD, MS, PhD

Academic Editor

Thank you for addressing all issues raised by reviewers.